# Autophagy and Its Relationship to Epithelial to Mesenchymal Transition: When Autophagy Inhibition for Cancer Therapy Turns Counterproductive

**DOI:** 10.3390/biology8040071

**Published:** 2019-09-24

**Authors:** Guadalupe Rojas-Sanchez, Israel Cotzomi-Ortega, Nidia G. Pazos-Salazar, Julio Reyes-Leyva, Paola Maycotte

**Affiliations:** 1Facultad de Ciencias Químicas, Benemérita Universidad Autónoma de Puebla, Ciudad Universitaria, Puebla 72570, Mexico; grs.pharm@gmail.com (G.R.-S.); israelcotzomi@gmail.com (I.C.-O.); nidianaye@hotmail.com (N.G.P.-S.); 2Centro de Investigación Biomédica de Oriente (CIBIOR), Instituto Mexicano del Seguro Social (IMSS), Km 4.5 Carretera Atlixco-Metepec HGZ5, Puebla 74360, Mexico; julio.reyes@imss.gob.mx; 3Consejo Nacional de Ciencia y Tecnología (CONACYT)—CIBIOR, IMSS, Puebla 74360, Mexico

**Keywords:** autophagy, cancer, cancer therapy, EMT

## Abstract

The manipulation of autophagy for cancer therapy has gained recent interest in clinical settings. Although inhibition of autophagy is currently being used in clinical trials for the treatment of several malignancies, autophagy has been shown to have diverse implications for normal cell homeostasis, cancer cell survival, and signaling to cells in the tumor microenvironment. Among these implications and of relevance for cancer therapy, the autophagic process is known to be involved in the regulation of protein secretion, in tumor cell immunogenicity, and in the regulation of epithelial-to-mesenchymal transition (EMT), a critical step in the process of cancer cell invasion. In this work, we have reviewed recent evidence linking autophagy to the regulation of EMT in cancer and normal epithelial cells, and have discussed important implications for the manipulation of autophagy during cancer therapy.

## 1. Introduction

Autophagy is a catabolic process occurring continually in eukaryotic cells at basal levels in which damaged or long-lived protein aggregates, organelles, and lipids are degraded. This process can also be induced to higher levels by several stress stimuli, such as nutrient deprivation, hypoxia, DNA damage, endoplasmic reticulum stress, cytotoxicity, and pathogens [1]. In cancer, autophagy has been described as a double-edged sword, since it is known to function as an antitumorigenic process in healthy cells, but to enhance tumor progression once a tumor is established [2]. Additionally, autophagy has been shown to be required for cancer cell survival to metabolic stress within the tumor and for cancer cell survival to therapy, and some tumor cells with particular mutations are known to be dependent on autophagy for survival, highlighting the need to incorporate this concept into clinical trial design [3] and underscoring the need for careful selection of the cancer types or of patients in which autophagy therapies should be used. In this regard, autophagy has also been implicated in protein secretion [4], in the regulation of immunogenicity [5], and in tumor cell invasion. Autophagy, or its inhibition, has been shown to regulate the secretion of proteins involved in tumorigenesis, survival, proliferation, tumor editing, and invasion [4] and has also been shown to be closely related to the regulation of epithelial-to-mesenchymal transition (EMT), one of the initial steps involved in the metastatic process [6,7].

Since several current clinical trials are exploring the inhibition of autophagy using pharmacological agents in combination with cancer therapies in diverse types of cancer [2], it is important to highlight the possible undesirable side-effects of autophagy manipulation for cancer therapy. In many of the clinical trials wherein autophagy inhibition was studied, patients have not always been selected based on autophagy-dependency biomarkers [3]. Thus, encouraging results from clinical trials could have been masked by cancer subtypes in which autophagy inhibition was not having an effect, or even worse, was promoting undesirable consequences for patient outcome. In this review, we have analyzed the evidence that autophagy inhibition by itself, or in combination with other cancer therapies, has been shown to promote malignancy in several types of cancer, with an emphasis on the promotion of EMT, suggesting that careful selection of the patients and cancer types in which autophagy should be manipulated is needed for an optimal targeting of autophagy for cancer therapy.

## 2. The Autophagic Pathway

Since it was first characterized, the autophagic process has been widely studied, and three different types of autophagy have been described depending on the route of protein or cargo delivery to the lysosome: macroautophagy, microautophagy, and chaperone-mediated autophagy [8]. Macroautophagy (hereafter referred to as autophagy) is perhaps the best characterized type of autophagy, with important implications in health and disease. It is a cellular housekeeping process by which cells eliminate damaged or long-lived proteins, lipids, and organelles via their sequestration in a de novo formed double membrane structure (phagophore), which subsequently closes (autophagosome) and is finally degraded after fusion with the lysosome (autolysosome) [8]. This cellular process is active at low levels in all cell types, but can be triggered by amino acid starvation, withdrawal of growth factors, low cellular energy status, hypoxia, pathogens, oxidative stress, genotoxic damage, and pathogens [9]. As autophagy has a pivotal role in cellular maintenance, its dysregulation has been related to chronic disease such as Parkinson’s or Alzheimer’s Diseases, congenital ataxia, cardiovascular diseases, aging, and cancer [10]. This clearance process is orchestrated by a set of well conserved autophagy-related (ATG) proteins from lower to higher eukaryotes [11,12]. In yeast, more than 40 ATG proteins have been described and are classified in a hierarchical fashion according to their function in the process [13,14]. Autophagy requires the termination of mammalian target of rapamycin complex 1 (mTORC1) signaling or activation of AMP-activated protein kinase (AMPK), which leads to ULK1/2 activation and ATG13 and FIP200 phosphorylation. This results in Beclin1 phosphorylation and its recruitment to the autophagosomal biogenesis membrane site together with VPS34, VPS15, and ATG14L to form the phosphatidylinositol 3-kinase (PI3K) complex, which produces phosphatidylinositol 3-phosphate (PI3P) [8,14,15]. PI3P is involved in membrane dynamics as well as ATG protein trafficking and recruitment to the autophagosomal membrane [15]. The PI3K complex is regulated by several Beclin1 interacting proteins, including BCL2, Rubicon, UVRAG-Bif-1, and AMBRA1 [8,15]. Proteins that take part in the elongation and closure process are also recruited to the outer surface of the isolation membrane, where two ubiquitin-like systems are activated [8,14]. In the first one, the ATG12–ATG5–ATG16L1 complex is formed by the sequential action of the ATG4 protease, the E1-like enzyme ATG7, and the E2-like enzyme ATG3. The complex then contributes to the conjugation of ATG8/LC3 to phosphatidylethanolamine (PE), producing LC3-II. LC3-II associates to both the inner and outer membrane of the autophagosome, where it can tether autophagic cargo through binding to specific adaptor proteins like p62/SQSTM1, NBR1, NDP52, OPTN, or TAX1BP1 at their LC3-interacting regions (LIR) [16]. p62 binds cytosolic ubiquinated protein aggregates or ubiquitinated proteins localized to organelle membranes such as those of mitochondria and peroxisomes for their degradation by mitophagy or pexophagy. The other LIR-containing proteins have been implicated in xenophagy (autophagic degradation of microorganisms), specific signaling pathways, or similar functions to p62 [16,17]. Once the autophagosomal membrane is closed, the autophagosome moves along microtubules to fuse with the lysosome, becoming an autolysosome, where cargo is degraded and recycled [8]. Importantly, LC3-II levels or LC3 punctate localization in the cytoplasm are currently used as a markers of autophagosome formation, and are the assays most widely used to monitor autophagy [18]. Some of the signaling pathways that modulate autophagy, as well as pharmacological modulators of the process, are described in Figure 1.

## 3. Autophagy in Cancer

The autophagic process has a complex and context-dependent role in cancer (Figure 2). Under basal conditions, autophagy has been shown to act as a tumor-suppressing mechanism in normal cells. In this regard, *Becn1* heterozygous mice have been shown to develop normally but with an increased frequency of lymphomas and carcinomas of the liver and lung [19]. Similar results have been observed in mice with systemic mosaic *Atg5* deletion or liver-specific deletion of *Atg7*, which led to the development of benign liver adenomas. In this work, autophagy-deficient hepatocytes presented mitochondrial swelling, DNA oxidative damage, and p62 accumulation. Importantly, tumor progression decreased after concomitant deletion of p62 in the *Atg7*^−/−^ background, indicating that p62 accumulation occurring due to autophagy inhibition played a major role in the promotion of tumorigenesis. Small tumors were still detected in the *Atg7*^flox/flox^, Alb-Cre, *p62*^−/−^ mouse liver, probably suggesting that p62 accumulation is important for tumor progression, but not for the initial transformation step [20]. The pro-tumorigenic role of p62 has been attributed to its activation of the Nrf2 pathway or to the deregulation of NF-κB signaling [20,21].

In a different setting, in the presence of a cancer driver stimulus, deficiency in autophagy caused by the genetic deletion of *Atg* genes has been shown to promote pre-malignant lesions in different mouse models [19]. Notably, *Becn1* heterozygous mice developing tumors have been found to retain the second allele of *Becn1* and to maintain functional autophagy [19]. Additionally, the core autophagy genes have not been found to be mutated in different types of cancer, and the majority of human cancers have been shown to have a functional, intact autophagic pathway, which has even been found in some cases to be transcriptionally up-regulated [22]. Thus, although decreased autophagy in normal cells would induce cellular damage that could lead to malignancy, a functional autophagic pathway is required for oncogenic progression. This has been demonstrated in diverse genetically modified cancer mouse models. In a pancreatic ductal adenocarcinoma mouse model with oncogenic *Kras*, autophagy inhibition increased the frequency of low-grade, pre-malignant pancreatic intraepithelial neoplasia formation, but blocked the progression to high grade intraepithelial neoplasia or adenocarcinoma [23]. In a KRas-driven model of lung cancer, *Atg5* deletion increased hyperplastic tumor foci formation, but decreased progression to adenocarcinomas and signs of malignancy [24]; in a *Kras, p53*^−/−^ mouse model of lung cancer, *Atg7* deletion altered tumor fate from adenomas to more benign oncocytomas, characterized by the accumulation of defective mitochondria [25]. With regards to tumor progression, autophagy has also been shown to be important in mediating survival to anoikis, a type of apoptosis mediated by substrate detachment and of which avoidance is necessary for tumor cell migration and invasion [26]. The abovementioned studies, together with the fact that autophagy regulates tumor cell survival by providing substrates to maintain rapidly multiplying and metabolically stressed tumor cells [27,28], suggest that targeting autophagy could be a therapeutic approach for cancer. 

Cancer patients are treated with surgery together with adjuvant or neoadjuvant therapies, which include radiation, cytotoxic chemotherapy, targeted therapies (in case that the oncogenic driver has been identified), or immunogenic therapy. An important role for autophagy has been described for many types of cancer and for the different types of cancer therapies, indicating a promising role for the manipulation of this process in clinical trials. Currently, several clinical trials are trying to inhibit autophagy in several types of cancer using chloroquine (CQ) or hydroxychloroquine (HCQ) alone or in combination with chemotherapy or targeted therapies [3]. 

In this regard, some oncogenic backgrounds have been associated with increased dependency on autophagy even in the absence of stress. This addiction to autophagy has been described for several tumors with driver mutations in the RAS/MAPK pathway, including pancreatic, lung, melanoma, brain, and colorectal cancers [3,22]. Other tumor mutations have also been proposed to be important for autophagy addiction or dependence, like alterations in the p53 pathway and activation of the STAT3 or EGFR pathways [3]. Importantly, autophagy has also been shown to mediate the acquisition of resistance to targeted therapies [29] and to avoid apoptosis through the degradation of specific pro-apoptotic stimuli [30], and has also been involved in other mechanisms known to promote malignancy, like the maintenance of cancer stem cells (CSCs), resistance to chemotherapy, and secretion of pro-inflammatory cytokines and matrix metalloproteinases (MMPs) [31,32,33], further supporting the rationale for the use of autophagy inhibitors for the treatment of cancer.

Besides the extensive scientific evidence and clinical trials indicating that autophagy should be inhibited during cancer therapy, there is also evidence in the literature that suggests that autophagy inhibition could promote tumorigenesis, invasion, and immunoediting (Table 1). In this regard, it has been proposed that autophagy is necessary for the immunogenicity of cell death. Autophagy has been proposed to increase the secretion of ATP in cells treated with immunogenic therapies, thus suggesting that autophagy inhibition would decrease the efficacy of immunogenic therapies [34]; it has also been suggested that autophagy inhibition might induce the expression of PD-L1 in tumor cells [35], indicating that the inhibition of autophagy could have a detrimental effect on cancer therapy, particularly in those treatments where the anti-tumoral immune response has an important role. Another important side-effect of the inhibition of autophagy is the promotion of EMT. Some of this evidence will be discussed in the following sections, since a precise understanding of the types of cancer where the inhibition of autophagy could promote invasion and metastasis is needed for the successful manipulation of autophagy during cancer therapy.

## 4. Epithelial to Mesenchymal Transition in Cancer

Cancer metastasis is a multi-step process in which epithelial primary tumors lose attachment to other cells and the basal membrane, acquire the ability to disrupt the basal membrane, extravasate, migrate to a target tissue, intravasate, and colonize. These sequential steps can be achieved through the activation of the EMT process, which is highly conserved among higher eukaryotes and is necessary for embryogenesis and wound healing [67,68,69]. Cancer cells hijack this process, allowing them to lose epithelial features such as apico-basal polarity and disrupt cell–cell junctions caused by the remodeling of the cytoskeleton, tight junctions, and hemidesmosomes, while undergoing a concomitant gain of mesenchymal characteristics like front–rear polarity, spindle-like shape, motility, invasiveness, stem-like features, immune escape markers, and chemoresistance [67,68,69,70]. Thus, in carcinomas, or cancer from epithelial tissues, EMT drives the invasion-–metastasis cascade. The inverse process, mesenchymal-to-epithelial transition (MET) has been proposed to be necessary for the establishment of at least some types of metastases, since epithelial markers have been found in some, but not all metastatic foci [71], and it has been proposed that MET defines metastatic organotropism [72].

During EMT, cells downregulate the expression of epithelial proteins, including those of cell junction complexes (E-cadherin, claudins, occludins, desmoplakin, and plakophilin), and redirect their gene expression patterns to promote changes in cytoskeletal architecture and promote adhesion to mesenchymal cells and the extracellular matrix through the expression of N-cadherin [69]. The EMT process is orchestrated by a set of EMT transcription factors (EMT-TFs) [69,73] such as SNAIL (SNAIL and SLUG), zinc-finger E-box binding (ZEB 1/2), and basic helix–loop–helix (E12, E47, TWIST1/2, ID) transcription factors [69]. EMT-TFs have in common the capacity to recognize E-boxes in target genes, including E-cadherin, and have also been reported to be involved in the control of apoptosis and stemness [74]. The expression of EMT-TFs is regulated at different levels, including epigenetic regulation, alternative splicing, regulation by miRNAs, or altered protein degradation and is mediated by distinct microenvironmental factors such as hypoxia or EMT-inducing cytokines secreted from the stroma [69,71].

Although it is generally accepted that aberrant activation of one or more EMT-TFs is necessary for establishment of metastasis, EMT-inducing pathways may vary between distinct cell types. For example, non-stem basal breast cancer cells lines have been found to have a chromatin configuration at the ZEB1 promoter poised (activating and repressing methylation patterns) to respond to TGFβ and rapidly induce ZEB1 expression, while luminal non-stem breast cancer cell lines were found to only have silencing methylation patterns [75], indicating different EMT processes or features depending on the cancer subtype, differential response to EMT-inducing cytokines, distinct EMT-TF phenotypes, and/or the activation of different migration strategies in response to components of the extracellular matrix [76].

It was generally accepted until recently that EMT was a binary process because cells could only have two fates: being epithelial or mesenchymal. However, this model has been recently challenged by the description of partial EMT phenotypes, where migrating cells do not completely lose their epithelial markers, indicating that morphological changes associated with EMT can occur even without a complete loss of an epithelial phenotype, and without the requirement of an abrupt upregulation of mesenchymal markers [77,78,79]. In this regard, in a lineage-traced *K-ras*^G12D^, *p53*^+/−^ pancreatic adenocarcinoma mouse model, different EMT phenotypes were dissected [77]: complete EMT involved the transcriptional repression of epithelial genes and the activation of a mesenchymal process [77,80], while partial EMT involved the re-localization of E-cadherin to intracellular vesicles without the transcriptional repression of the gene. Importantly, complete EMT was associated with a single cell invasion profile and correlated with basal-like cancer subtypes of diverse tissues. On the other hand, partial EMT was associated with a collective dissemination profile in which E-cadherin was found to be expressed at the cell–cell contact sites of the migrating cluster. The partial EMT gene signature correlated with well differentiated subtypes of cancer, and both mechanisms were found to be conserved in cell lines of other cancers, such as breast and colorectal cancer cell lines [77]. Importantly, in circulating tumor cells (CTCs) of patients with esophageal squamous cell carcinoma [81], invasive breast cancer [82], and hepatocellular carcinoma [83], partial EMT has been observed and correlated with aggressiveness and poor clinical outcomes, and in a metastasis breast cancer model, CTCs with epithelial markers and a restricted mesenchymal transition had the strongest lung metastasis formation ability [79].

Multiple signaling pathways cooperate in the initiation and progression of EMT, including TGFβ, WNT family proteins, Notch, HIF1α, and growth factors that act through the RTKs/MAPK pathway. These pathways often converge in the activation of EMT-TFs, and are initially activated by extracellular cues [69]. In this regard, growth factors and cytokines secreted by neighboring cells or cells from the microenvironment have been characterized as inducing EMT in cancer cells. Thus, EGF, IGF1, FGF, HGF, or PDGF, which are normally found in the tumor microenvironment, signal EMT through their receptors in cancer cells. Hypoxia in tumor cells is also known to activate HIF1α and favor EMT in hypoxic tumor cells. Aberrant WNT or Hedgehog signaling in cancer cells are also known to activate EMT signaling, and probably the best characterized EMT inducer is transforming growth factor β (TGFβ), which is known to induce EMT through the activation of SMAD, PI3K/AKT, MAPK, and RHO-GTPases [69]. 

Besides TGFβ, pro-inflammatory cytokines, either secreted by cancer cells or by immune cells in the tumor microenvironment, have been shown to have an important effect on the regulation of EMT [84,85]. In the following section, we briefly discuss the role of cytokines and other soluble factors known to be present in the tumor microenvironment, which are able to activate and maintain EMT.

## 5. Cytokines in the Tumor Microenvironment and Their Effects on EMT

Tumors are complex structures comprised of cancer cells and other cell types which are recruited to the tumor site and influenced by their presence in the tumor. Non-malignant cells in the tumor microenvironment include cells of the immune system, cells from the tumor vasculature, and lymphatics, fibroblasts, pericytes, and adipocytes [86]. In this setting, inter-cellular communication creates a complex network of interactions between malignant and non-transformed cells, mostly mediated by extracellular protein secretion (cytokines, chemokines, growth factors, and inflammatory and matrix remodeling enzymes) and inter-cellular signaling (Figure 3). Immune cells in the tumor microenvironment include T and B lymphocytes, natural killer (NK and NKT) cells, tumor-associated macrophages (TAMs), myeloid-derived suppressor cells (MDSCs), dendritic cells, and tumor-associated neutrophils (TANs) [86], which, along with tumor cells, are the major source of cytokines and chemokines in the tumor microenvironment. 

Anti-tumor immunity presumably starts with the presentation of tumor antigens by tissue-resident dendritic cells (DCs) or by DCs in the draining lymph nodes to T cells in the lymphoid tissues. Upon activation, CD4^+^ T cells can give rise to Th cells with distinct cytokine profiles, or to regulatory cells (Treg). Th1 cells secrete IL-1, TNFα, and IFNγ and, in conjunction with cytotoxic CD8^+^ T cells, promote M1 macrophage polarization and cytotoxic activity. In contrast, Th2 cells, by secreting IL-4,-5,-6,-10, and -13, induce loss of T-cell-mediated cytotoxicity and enhance the tumor-promoting activities of macrophages (M2 polarization) [87].

The cells best characterized as a source of inflammatory mediators in the tumor microenvironment are TAMs, which are known to account for about 30 to 50% of the tumor mass [84] and thus, their recruitment and polarization to the tumor site has a dominant role in driving and maintaining cancer-related inflammation [88]. Cytokines and chemokines derived from tumor cells and cancer-associated fibroblasts (CAFs), such as CSF-1, CXCL12/SDF1, CCL2/MCP-1, CCL5/RANTES, and VEGF, recruit mononuclear cells into the tumor microenvironment and activate them to become TAMs or MDSCs [84,88]. CSF-1 drives TAM differentiation towards an immunosuppressive, tumor-promoting, M2-like phenotype, while IFNγ or GM-CSF promote classically activated M1 macrophages. M1 macrophages are known to have an important role in the immunoediting phase of early tumor elimination, while in established tumors, Th2 cytokines (IL-4 and IL-13) elicit alternative M2 activation [88]. TAMs have an important role in tumor progression. They are known to produce growth factors, such as EGF; proteolytic enzymes that digest the extracellular matrix; IL-1, which has an important role on tumor cell EMT; and, when present at distant sites, can provide a supportive niche for metastasis establishment [88]. Additionally, oxidation products generated by TAMs contribute to cancer cell genetic instability, and IL-10 and TGFβ promote the immunosuppressive activity of Treg cells and in general have immunosuppressive effects via prostaglandin production, supporting tumor progression [88]. Another immunosuppressive effect is mediated by IL-6 secreted from tumor cells, CAFs, and other tumor stromal cells which leads to MDSC recruitment to the tumor microenvironment [89,90]. Importantly, accumulation of MDSCs promotes and fuels chronic inflammation and immunosuppression through promotion of Treg and CAF differentiation and by their differentiation into TAMs [91].

Thus, cytokines and growth factors secreted by tumor cells, stromal cells, or cells from the immune system in the tumor niche have an important role in tumor maintenance and are known to regulate cell to cell communication in an autocrine and paracrine fashion to control proliferation, cell survival, death, immunoediting, angiogenesis, and cell migration of tumoral cells [92]. The immunological profile of cancer-related chronic inflammation thus includes both inflammatory and immunosuppressive cytokines such as IL-1β, IL-6, TGFβ, IL-10, and TNF [91], and their effects primarily induce the activation and maintenance of the JAK2/STAT3, RAS/MAPKs, NF-κB, Wnt, and PI3K/Akt pathways [93,94,95,96,97]. High levels of these cytokines have been found in serum from patients and mouse models from different types of cancers such as breast [98,99,100,101,102,103], colorectal [104], NSCLC [105,106] and head and neck cancer [107,108,109], ovarian cancer, and several hematological malignancies [100], and their increase has been found to correlate with malignancy and a bad prognosis.

Increased inflammation within the tumor microenvironment has been shown to have an important role in the induction of pro-tumorigenic signaling in cancer cells. For instance, IL-1 has been shown to promote cancer cell invasion, metastasis, chemoresistance, and maintenance of CSCs. IL-1, TGFβ, and TLR signaling have been shown to converge in TAK1/MAP3K activation, controlling the activation of different transcription factors such as AP-1 and NF-κB, inducing inflammation, proliferation, and chemoresistance in tumor cells [96]. An autocrine IL-1β signaling loop has also been described for CD133^+^ CSCs mediating NF-κB signaling, EMT, and invasion in pancreatic cancer cell lines [110], and exposure to IL-1β induced motility and invasiveness in a non-invasive breast cancer cell line [97].

IL-6 has also been shown to have diagnostic or prognostic relevance in several diseases, including cancer [111]. IL-6 family members signal through JAK2 and TYK2 and can activate the STAT, RAS/MAPK, AKT/PI3K, and NOTCH pathways, which activate proliferation, survival, migration, metabolism, and oxidative stress, and thus have been described as prototypical pro-tumorigenic cytokines [111]. Their expression and release is mediated by NF-κB and STAT3 [112], and is activated by other proinflammatory cytokines or by IL-6 itself. Indeed, exposure to IL-6 induced EMT-TF as well as vimentin and N-cadherin expression in non-invasive breast carcinoma cells [103], and induced STAT3-mediated expression of oncomirs miR-21, an antiapoptotic miRNA, and of miR-200, an EMT-related miRNA, in multiple myeloma [113] and gastric cancer [114], respectively. Moreover, IL-6 signaling (IL-6, LIF, or OSM) has been described to be important for CSC induction and maintenance in breast [115], ovarian [116], endometrial [117], and pancreatic cancers [118]. Other cytokines which are known to have an important effect on EMT and the promotion of malignancy are IL-8 [119], MCP-1 [120], and TGFβ.

As mentioned previously, TGFβ is an established inducer of EMT through activation of SMAD signaling and induction of EMT-TF [85]. TGFβ has also been shown to induce CSC marker expression in different types of cancers [84], and TNFα and IL-6 can synergistically activate the TGFβ signaling pathway through activation of NF-κB, the activation of which also induces the expression of EMT-TF and IL-6 secretion [69,85]. Due to the important role of cytokines in the regulation of the tumoral immune response, as well as on tumor cell oncogenic signaling, some cytokines have been approved for the treatment of human cancer and others are currently being explored in clinical trials [92,112,121]. Finally, and of relevance to the autophagy field, autophagy has been implicated in the secretion of several pro-inflammatory cytokines, particularly those involved in the promotion of EMT. Thus, as is further discussed in the following section, autophagy-mediated secretion and the secretion of cytokines mediated by the inhibition of autophagy are an important link between autophagy and the modulation of EMT.

## 6. Autophagy and Its Effects on Epithelial to Mesenchymal Transition in Cancer Cells

One of the first links between autophagy and the invasion process was the discovery that autophagy allows epithelial cells to survive anoikis [26]. Since then, autophagy has also been shown to be implicated in several aspects of the metastatic process, like promoting motility, inducing the degradation of the extracellular matrix by regulating protein secretion, pre-metastatic niche priming [122], metastasis establishment [31], and CSC maintenance [33] (Figure 2). Importantly, although considerable evidence suggests that the inhibition of autophagy will improve cancer therapy, particularly in those types of cancer which are dependent on autophagy, and despite early phase clinical trials showing promising results of the use of hydroxychloroquine for the inhibition of autophagy [123], other works have highlighted possible undesirable effects of the inhibition of autophagy during cancer therapy (Table 1). More recently, the role of autophagy in EMT has begun to be established, and controversies exist in the literature regarding the role of autophagy inhibition on EMT; while several studies implicate autophagy in the promotion of EMT, other works have suggested the inverse, indicating that at least in some cases, inhibition of autophagy could be promoting EMT and thus inducing cancer cell invasion (Table 1).

In agreement with the former idea and supporting the beneficial effect of the inhibition of autophagy during cancer therapy, several works have implicated autophagy in the promotion of EMT. For instance, autophagy has been shown to be necessary for EMT induction in a model of DRAM1 (a p53-mediated regulator of autophagy)-mediated EMT, invasion, and metastasis in hepatoblastoma cells [52], as well as for TGFβ2-induced EMT and reactive oxygen species (ROS) modulation in hepatocellular carcinoma [51], or for TGFβ1-induced EMT in non-small-cell lung carcinoma (NSCLC) cell lines [54]. Additionally, starvation-induced autophagy was shown to be responsible for phosphodiesterase 4A degradation, cAMP/PKA/CREB signaling [50], TGFβ1 expression, and EMT markers in hepatocellular carcinoma cells [49,50], and rapamycin induced migration, invasion, and EMT marker expression in colorectal cancer cells, which could be decreased by *beclin 1* knockdown [56]. mTOR signaling inhibition (which induces autophagy), attenuated migration and invasion, and decreased EMT marker expression in colorectal cancer cells [124]. While the abovementioned evidence supports an EMT-promoting role of autophagy, in another model of hepatocellular carcinoma, inhibition of autophagy had no effect on cell migration, invasion, or EMT marker expression in vitro, but sensitized cells to anoikis and thus decreased lung metastases [47], indicating that the role of autophagy in EMT is context-dependent and/or that the effects of the inhibition of autophagy in the establishment of metastasis are not necessarily due to its effects on EMT, but rather on its effects on other steps of the metastatic process or in the promotion of cell death.

On the other hand, recent evidence has also suggested the opposite, and studies in different types of cancers have indicated that the inhibition of autophagy could promote EMT. As an example, autophagy has been implicated in the degradation of EMT-TF. Autophagy has been shown to mediate p62-dependent Twist1 degradation in *Atg* KO MEFS or in human squamous carcinoma cells. In the same study, increased p62 was required for the maintenance of high Twist1 and decreased E-cadherin protein levels in EGF/TGFβ-induced EMT [39]. Autophagy also has been suggested to attenuate DEDD (death-effector domain-containing DNA-binding protein)-induced EMT by inducing the degradation of Snail and Twist in breast cancer cell lines [59] or by degrading Snail in lung or cervical cancer cell lines [61], or during TGFβ-induced EMT in immortalized hepatocytes in a p62-dependent manner [62]. Other EMT-promoting effects independent of the autophagic degradation of EMT-TF have been described for the inhibition of autophagy. Thus, genetic inhibition of autophagy has been shown to increase migration, invasiveness, and expression of EMT-TF (at the mRNA level, indicating this not due to EMT-TF protein degradation by autophagy) in RAS-transformed cancer cell lines. The mechanism described in this study involves p62 accumulation and NF-κB activation, since p62 knockdown decreased NF-κB reporter activity and p62 or RELA (an NF-kB subunit) knockdown decreased the expression of EMT-TF [53]. A similar mechanism has been described in gastric cancer cells, where *beclin 1* knockdown induced ROS-dependent NF-κB activation, HIF1α expression, and EMT [46]. Importantly, in the same study, antioxidant treatment reverted autophagy-inhibition-induced metastasis in vivo [46]. In ovarian cancer cell lines, autophagy inhibition increased ROS production, migration, invasion, and EMT marker expression. In this work, the mechanism proposed involves ROS-mediated expression of heme oxigenase-1 (HO-1), an NRF2, NF-κB, or AP2 target which has been associated with malignancy and invasion in different types of cancer [57]. Additionally, in a breast cancer xenograft mouse model, a hypoxia-inducible dominant negative ULK1 to block autophagy in the MDA-MB-231 breast cancer cell line did not have an effect on tumor formation, but increased metastasis to the lungs [48]. This study also found reduced autophagy-related gene expression, as well as increased p62 levels, to predict poor prognosis in breast cancer patients [48]. The abovementioned studies suggest an important role for ROS and/or p62 signaling pathway activation induced by the inhibition of autophagy, which could result in NRF2, NF-κB, and/or HIF1α activation and the promotion of EMT. Importantly, activation of these pathways by autophagy inhibition has also been involved in other malignancy-related features like expression of immune checkpoint inhibitor markers [35]. Another mechanism independent of EMT-TF by which autophagy inhibition could be promoting malignancy has been described, in which autophagy has been implicated in the degradation of the Notch1 intracellular domain, dependent on its interaction with p62. In this work, autophagy inhibition increased migration and invasiveness of MDA-MB-231 breast cancer cells, and this effect could be decreased with a Notch1 inhibitor [60]. Finally, in a thyroid cancer model, where cadherin-6 (CDH6) expression was associated with EMT and aggressiveness, mesenchymal features were related to increased mitochondrial fragmentation and decreased autophagy, while loss of CDH6 decreased mesenchymal features and induced mitochondrial elongation [125]. These findings suggest an intricate link between mitochondrial dynamics, autophagy, and cytoskeletal reorganization occurring during EMT, and highlight the need for understanding of how these processes regulate one another in order to effectively target them for inhibiting cancer progression [126].

Another important issue to be considered regarding the effects of the modulation of autophagy on EMT is the fact that autophagy has been implicated in protein secretion, particularly the secretion of pro-inflammatory cytokines [4]. In this regard, the autophagic pathway has been closely related to unconventional protein secretion pathways in a type of secretion termed secretory autophagy. Perhaps the best characterized example is the secretion of the proinflammatory cytokine IL-1β from mammalian cells. This secretion is dependent on inflammasome activation, and specialized secretory autophagosomes containing IL-1β and ferritin have been identified (Reviewed in Reference [4]). Autophagy has also been related to the secretion of IL-6 and IL-8, although the precise mechanism is not understood. *H-ras^V12^*-transformed fibroblasts had higher autophagy levels than their non-transformed controls and maintained senescence through IL-6 and IL-8 secretion [127]. Similarly, in *HRAS^V12^*-transformed MCF10A cells, genetic inhibition of autophagy decreased IL-6, MMP2, and 9 secretion and invasive characteristics [31]. Therefore, autophagy could have a proinflammatory function, since its inhibition prevents proinflammatory cytokine secretion. Whether this is a general mechanism regulating the secretion of IL-6 has been questioned, since in two breast cancer cell lines, genetic inhibition of autophagy decreased IL-6 secretion in triple negative cell lines, but increased IL-6 secretion in a luminal breast cancer cell line [32]. In the same line, melanoma cells had different secretion profiles depending on whether they had low or high autophagy levels. Cells with high autophagy secreted higher levels of IL-1β, LIF, CXCL8, and MMP2, all with known roles in inflammation and tumorigenesis [128], and secretion was downregulated by *ATG7* silencing, indicating that secretion was dependent on autophagy and that high autophagy levels could be used as a marker of highly secretory cells. In agreement with the previous data, skin cells of mice treated with UV light showed increased levels of autophagy and higher secretion of proinflammatory cytokines (such as CSF3/G-CSF, CXCL1, IL-6, TREM1, CCL2, CCL3/MIP-1α, IL-1β, and CXCL2), and secretion was related to the promotion of tumorigenesis [129].

Conversely, an anti-inflammatory role for autophagy has also been proposed in autophagy-deficient cancer mouse models. In a breast cancer model, *FIP200* deletion decreased tumorigenesis and metastasis. *FIP200* deletion induced an anti-tumor immune response mediated by CD8^+^ and CD4^+^ T cell infiltration to the tumor site, mediated by IFNγ and chemokine secretion (CXCL9, CXCL10, CXCL11) by autophagy-deficient tumor cells [40]. Additionally, in mice with *Kras*-driven lung tumors, *Atg7* deficiency reduced tumor burden but developed macrophage and lymphocyte infiltration and an extreme inflammatory response. Importantly, inflammation did not occur in the absence of *p53*, suggesting an important role for this tumor suppressor gene in the promotion of inflammation mediated by the inhibition of autophagy [25]. In agreement with the anti-inflammatory role of autophagy, pro-inflammatory cytokines (IL-1β, IL-18, and MIF) are also known to be secreted as a consequence of autophagy inhibition. The mechanism proposed is the accumulation of damaged mitochondria due to decreased mitophagy, which would induce excessive ROS production and inflammasome activation, leading to interleukin activation and secretion (Reviewed in Reference [4]). 

Due to the important role of cytokines in the regulation of EMT and tumor promotion, it is imperative to understand how the manipulation of autophagy is influencing protein secretion, and in which cell types this manipulation could have an unfavorable outcome for cancer therapy. 

## 7. Final Remarks

There is extensive evidence in the literature and promising results from cancer patient clinical trials to suggest that the inhibition of autophagy using chloroquine or hydroxychloroquine is safe and has been well tolerated by cancer patients. Additionally, published phase I and II clinical trials using autophagy inhibitors along with chemotherapy or targeted therapies strongly suggest that this approach will have a favorable outcome for cancer therapy [3,123,130]. Importantly, the best response is expected in appropriately selected patient groups, since evidence suggests that autophagy should be inhibited particularly in those cancers that are the most sensitive to autophagy inhibition or the so-called autophagy-dependent cancers, which have activating mutations in the EGFR/RAS/BRAF signaling pathway [3,123]. Clinical evidence suggests that another group of patients that will benefit by the addition of autophagy inhibitors to their therapeutic regime will be those whose cancers have developed resistance to targeted therapy, as has been shown for BRAF-inhibitor-resistant brain tumors [3,29] or those receiving chemotherapies which have been shown to induce protective autophagy in cancer cells, as has been shown for the addition of HCQ to patients with advanced solid tumors and melanoma treated with temozolomide [131].

Despite encouraging data from patient clinical trials, the fact that the inhibition of autophagy could have undesirable effects in some types of cancer and, as we have discussed in this manuscript, could induce EMT and promote invasion and metastasis, at least in some cases, is a worrisome finding. The mechanisms by which autophagy inhibition would induce EMT include EMT-TF degradation by autophagy, or the accumulation of p62 due to the inhibition of autophagy and a consequent increase in NF-κB signaling and transcriptional induction of EMT-TF. If these are the main mechanisms by which inhibition of autophagy promotes EMT, it will be necessary to investigate whether autophagic degradation of EMT-TF is a general mechanism or if it only occurs in certain cell types, and whether accumulation of p62 and its induction of NF-κB signaling occurs in certain cell types but not others when autophagy is inhibited. Another mechanism proposed for the promotion of malignancy mediated by the inhibition of autophagy is the degradation of the Notch1 intracellular domain by autophagy [60]. Thus, it will be important to identify those cancers where autophagy is constitutively inhibiting Notch signaling, and either avoid using autophagy inhibitors or use them in combination with inhibitors of Notch. 

An interesting approach for targeting EMT induced by the inhibition of autophagy has been suggested by Wang et al. [53], in the best characterized model of autophagy addiction, i.e., RAS-mutated cancer cells. In this study, despite displaying reduced tumor growth, *HRas^V12^* expressing *atg5^−/−^* or *atg7^−/−^* tumors had increased EMT marker expression, which was reverted using a NF-κB inhibitor. In this regard, many open clinical trials for targeting autophagy-dependent cancers are using RAS/MAPK mutation or activation as a biomarker for autophagy dependency, and these findings suggest avoiding the use of CQ or HCQ as single agents, or using them along with a NF-κB inhibitor or inhibitors of malignancy-associated signaling pathways that could be activated by the inhibition of autophagy, even in cancers where the best outcomes are expected. Additionally, since autophagy regulates cytokine secretion in cancer cells and in other cell types, and since cytokine secretion has a leading effect on the promotion of EMT, it will be imperative to identify those cancer cells or cells from the immune system where the inhibition of autophagy promotes pro-tumorigenic cytokine secretion, as well as the contribution of each cell type to the cytokine pool in the tumor microenvironment after the inhibition of autophagy.

Fortunately, no unfavorable outcomes have been reported for the clinical trials using inhibitors of autophagy for the treatment of several types of cancer. With this review, we do not intend to discourage the use of autophagy inhibitors in cancer clinical trials, but rather to discuss the evidence in the literature, mostly in vitro but also in vivo, that suggests that autophagy inhibition could be promoting EMT in at least some cases. Thus, it will be central to understand in which cancer types autophagy inhibition could have a counter-productive effect, and, if blocking autophagy could promote invasion, if this would be particular to a certain cancer type or if this effect is likely to be more evident in early-stage malignancies that retain their epithelial features and have not triggered EMT.

## Figures and Tables

**Figure 1 biology-08-00071-f001:**
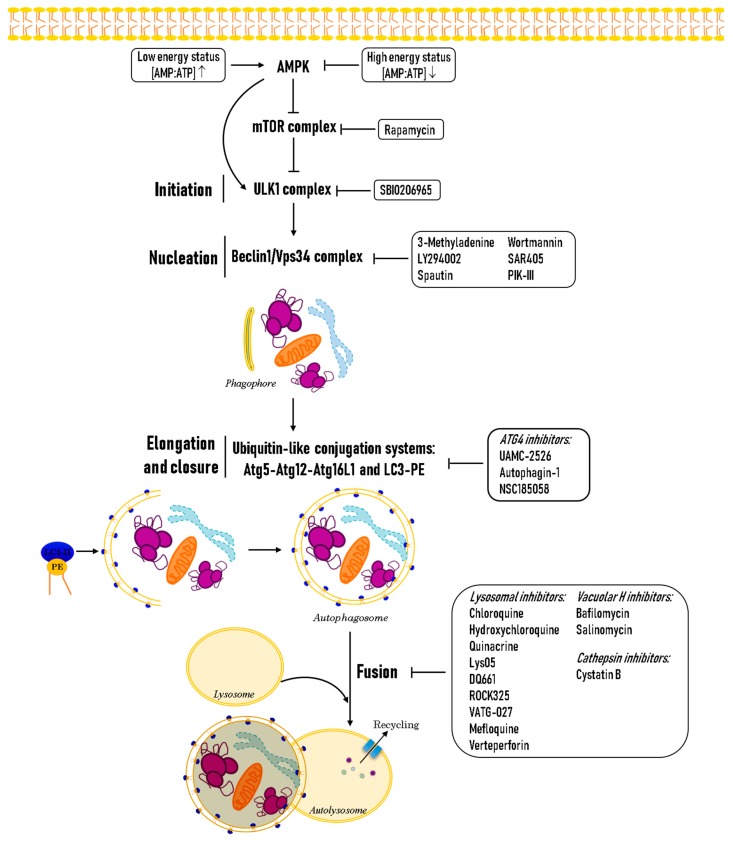
The autophagic pathway and its regulation. Two major regulators of autophagy are the mammalian target of rapamycin complex 1 (mTORC1) and AMP-activated protein kinase (AMPK). In amino-acid-rich conditions, mTORC1 negatively modulates autophagy. Under nutrient deprivation or low energy levels (sensed by AMPK), mTORC1 is inhibited and autophagy is induced. Growth factor withdrawal and hypoxia are also known triggers of autophagy. Pharmacological inhibitors of the different steps of the autophagic pathway have been described. It is of clinical relevance, since they are FDA approved drugs, that chloroquine and hydroxychloroquine are the only drugs currently used in clinical trials to inhibit autophagy in solid tumors.

**Figure 2 biology-08-00071-f002:**
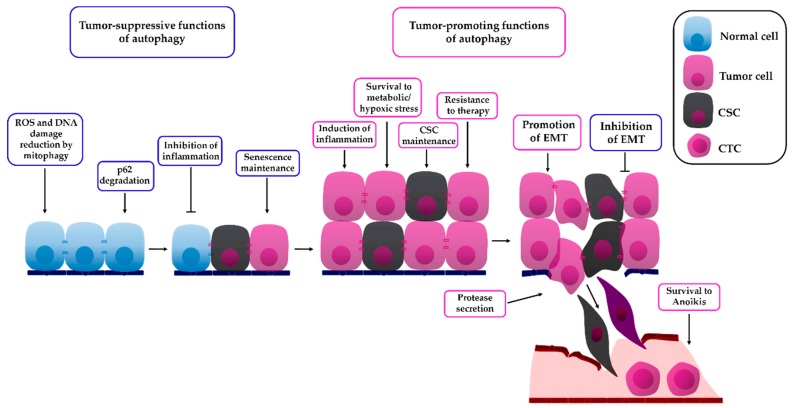
The role of autophagy in cancer. Autophagy functions as a tumor-suppressing process in normal cells by removing damaged proteins and organelles, maintaining low reactive oxygen species (ROS) levels by mitochondria elimination and decreasing DNA damage and genome instability. Autophagy also degrades p62, the accumulation of which is known to induce pro-tumorigenic signaling and induce inflammation. Once transformation has occurred, autophagy can also maintain cellular senescence to avoid the proliferation of transformed cells. On the other hand, once a tumor is established, tumor cells use autophagy as a survival mechanism to survive metabolic stress and hypoxia, to maintain tumor-related inflammation, to maintain cancer stem cell (CSC) survival, and to survive to cancer therapy. Additionally, during the invasion process, autophagy has been shown to be necessary for metalloprotease secretion, for degradation of the extracellular matrix, and for survival to anoikis in circulating tumor cells (CTCs). Evidence suggests that autophagy could also have a dual role in the regulation of EMT, and that inhibition of autophagy might be beneficial for some patients, since it would eliminate autophagy-dependent tumor cells and avoid invasion.

**Figure 3 biology-08-00071-f003:**
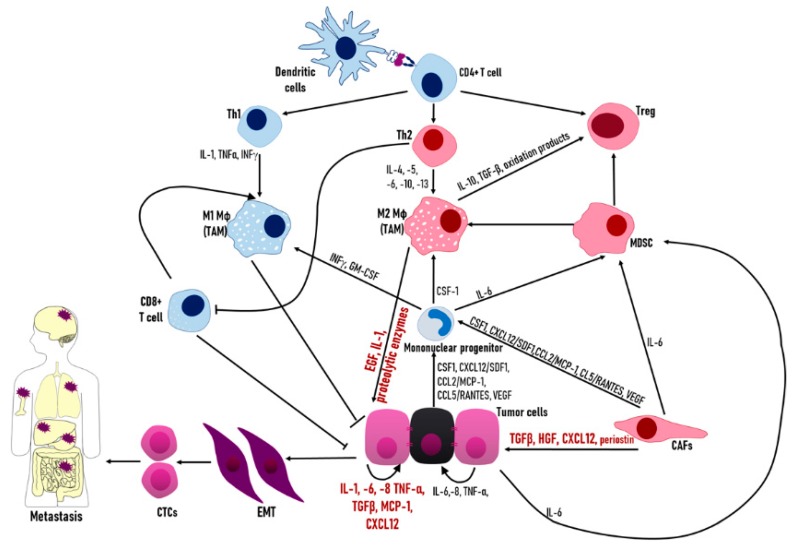
Cytokine signaling in the tumor microenvironment. Tumor cells can control the tumor microenvironment by secreting cytokines and chemokines, which recruit and define the type of immune cells in the tumor, as well as inducing the formation of cancer associated fibroblasts (CAFs). Cytokine secretion can also influence malignancy in neighboring tumor cells by the induction of EMT and by inducing the formation of circulating tumor cells (CTCs) and metastasis. In advanced stages of cancer, a complex communication network among tumor cells, immune cells, and the other cells in the tumor microenvironment defines immune evasion, and induces proliferation and malignancy. In the figure, cytokines, chemokines, and growth factors that promote epithelial to mesenchymal transition (EMT) and malignancy are marked in red. Anti-tumorigenic immune cells are marked in blue and immune cells associated with tumor promotion are marked in red. Cancer cells are marked in purple and CSC in black.

**Table 1 biology-08-00071-t001:** Beneficial and undesirable effects of the inhibition of autophagy in the treatment of cancer. Despite extensive evidence showing the potential for autophagy inhibition during cancer therapy, other studies suggest that inhibition of autophagy could have undesirable effects during cancer therapy. Atg, autophagy-related; MMTV-PyMT, mouse mammary tumor virus promoter/enhancer-polyomavirus middle T-antigen; MEFs, mouse embryonic fibroblasts; KO, knockout; CRC, colorectal cancer; CSCs, cancer stem cells; HCC, hepatocellular carcinoma; EMT, epithelial-to-mesenchymal transition.

Beneficial Effects of Autophagy Inhibition	Cancer-Related Feature	Counter-Productive Effects of Autophagy Inhibition
RAS-transformed cancer cells [36,37]	Proliferation/cancer Progression	
*K-ras^G12D^* or *Braf^V600E^;atg5/7^flox/flox^* mouse lung cancer models [24,25,38]	MEFs with Atg gene KO; epidermal squamous cell carcinoma mouse xenografts [39]
MMTV-PyMT mice with FIP200^−/−^ in mammary epithelial cells [40]	Dying autophagy-deficient cell lines induced proliferation of resistant cells in response to targeted therapy [41]
KRAS mutant pancreatic cancer [42]	Mice with autophagy inhibition together with p53^−/−^ had increased pancreatic ductal adenocarcinoma frequency [23]
BRAF^V600E^ central nervous system tumors [43]	
Triple negative breast cancer cell lines [44]	
RACK1-induced autophagy in CRC cells [45]		
RAS transformed cancer cells [37]	Migration/invasion/EMT or metastasis establishment	Gastric cancer cell lines and mouse xenografts [46]
HCC cell lines and xenografts [47]	Loss of ULK1 to suppress autophagy in the MDA-MB-231 breast cancer cell line during hypoxia [48]
Starvation- [49,50], TGF-β2- [51], or DRAM1-induced [52] autophagy in hepatic carcinoma cell lines	RAS-mutated cancer cells [53]
TGFβ1- or rapamycin-induced autophagy in non-small cell lung cancer cells [54]	Glioblastoma cell lines [55]
Rapamycin-induced autophagy in CRC cell lines [56]	Ovarian cancer cell lines [57]
Cisplatin-induced autophagy in nasopharyngeal carcinoma cells [58]	DEDD-induced autophagy in breast cancer cell lines [59]
	MEFs with *Atg* gene KO; epidermal squamous cell carcinoma mouse xenografts [39]
	MDA-MB-231 breast cancer cell line [60]
	H1299 lung or HeLa cervical cancer cell lines [61]
	Liver-specific autophagy-deficiency or TGFβ−treated immortalized hepatocytes [62]
MMTV-PyMT mice with *FIP200*^−/−^ conditional KO in mammary epithelial cells [40]	Immunoediting	Gastric cancer cell lines [35]
*K-ras^G12D^;Atg5^flox/flox^* mouse lung cancer model [24]	Colorectal or osteosarcoma cancer cell lines treated with immunogenic chemotherapy [34]
Ovarian cancer spheroids [63]	Tumor-initiating cells/CSCs	
Breast cancer stem cells [32,64,65]	
Hepatic cancer stem cells [66]

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
