# Peer review of "Autophagy and Its Relationship to Epithelial to Mesenchymal Transition: When Autophagy Inhibition for Cancer Therapy Turns Counterproductive"

_biology, 2019, doi:10.3390/biology8040071_

Round 1

Reviewer 1 Report

In this manuscript, the authors reviewed evidence in support of the claims that inhibition of autophagy as a cancer therapy can be counter-productive. That is, in certain patients and in certain cancer subtypes, the inhibition of autophagy may promote epithelial-to-mesenchymal-transition (EMT). The authors started the manuscript by introducing both autophagy and EMT. Then they described the role of each in tumor development. Finally, they summarized the literatures on the link between autophagy and EMT in the context of cancer. Moreover, the authors suggested that careful case selection, be it patients or cancer subtypes, for treatments including autophagy inhibition can remedy the undesired effects.

Although the article came on one side of the argument, it presented well balanced review of the issue. In particular, studies on the beneficiary/counterproductive roles of autophagy/autophagy inhibition were presented. Because the article suggests that autophagy inhibition therapies on trials may carry the risk of promoting tumor progression, it would be fitting to present findings, if available, from completed or ongoing trails to support the claim.

The risks of autophagy inhibition therapies have been previously suggested and this review is a valuable contribution on the topic especially by attributing them to promoting EMT. These therapies seem to be working in certain contexts. So, the authors may need to comment on as to why this is the case in some but not other cancer types. In addition, alternative explanations rather than the tumor specificity may be discussed.

The authors went into enough details about basic autophagy and EMT concepts. In addition, they used detailed infographics to simplify complex pathways and mechanisms. A brief introduction to the cancer therapies in general or the ones referred and used in combination with autophagy inhibitors would increase the accessibility of the review to the reader.

The authors described autophagy-dependent degradation of some EMT or metastasis-related proteins such as Snail, Twist, NICD1 so on. In addition, they need to discuss in detail these control pathways described in the following references;

- Zada et al. Control of the epithelial-to-mesenchymal transition and cancer metastasis by autophagy-dependent snai1 degradation. Cells. 2019 ;8.129. doi: 10.3390/cells8020129.

- Qiang, L.; He, Y.-Y. Autophagy deficiency stabilizes TWIST1 to promote epithelial- mesenchymal transition. Autophagy 2014, 10, 1864–5, doi:10.4161/auto.32171.

- Gugnoni, M.; Sancisi, V.; Manzotti, G.; Gandolfi, G.; Ciarrocchi, A. Autophagy and epithelial-mesenchymal transition: an intricate interplay in cancer. Cell Death Dis. 2016, 7, e2520, doi:10.1038/cddis.2016.415.

- Qiang, L.; He, Y.-Y. Autophagy deficiency stabilizes TWIST1 to promote epithelial- mesenchymal transition. Autophagy 2014, 10, 1864–5, doi:10.4161/auto.32171.

- Grassi, G.; Di Caprio, G.; Santangelo, L.; Fimia, G. M.; Cozzolino, A. M.; Komatsu, M.; Ippolito, G.; Tripodi, M.; Alonzi, T. Autophagy regulates hepatocyte identity and epithelial-to-mesenchymal and mesenchymal-to-epithelial transitions promoting Snail degradation. Cell Death Dis. 2015, 6, e1880, doi:10.1038/cddis.2015.249

- Gugnoni, M.; Sancisi, V.; Gandolfi, G.; Manzotti, G.; Ragazzi, M.; Giordano, D.; Tamagnini, I. Tigano, M.; Frasoldati, A.; Piana, S.; Ciarrocchi, A. Cadherin-6 promotes EMT and cancer metastasis by restraining autophagy. Oncogene 2017, 36, 667–677. doi:10.1038/onc.2016.237.

- Ahn, J.-S.; Ann, E.-J.; Kim, M.-Y.; Yoon, J.-H.; Lee, H.-J.; Jo, E.-H.; Lee, K.; Lee, J. S.; Park H.-S. Autophagy negatively regulates tumor cell proliferation through phosphorylation. dependent degradation of the Notch1 intracellular domain. Oncotarget 2016, 7, 79047-79063, doi:10.18632/oncotarget.12986.

Minor points

Each figure legends repeat some sentences described in the main text. Authors described the effects of cytokines on EMT in many pages, but they did not mention the functional connection of autophagy with those cytokines during cancer progression. Please cross check references. In page 5, the sentence from line 133 to 138 is not clear. There are some typos

Author Response

On behalf of my co-authors, I would like to thank you for careful revision of our manuscript and your insightful comments that we consider have significantly improved our paper. Below are our responses to your comments marked in red.

In this manuscript, the authors reviewed evidence in support of the claims that inhibition of autophagy as a cancer therapy can be counter-productive. That is, in certain patients and in certain cancer subtypes, the inhibition of autophagy may promote epithelial-to-mesenchymal-transition (EMT). The authors started the manuscript by introducing both autophagy and EMT. Then they described the role of each in tumor development. Finally, they summarized the literatures on the link between autophagy and EMT in the context of cancer. Moreover, the authors suggested that careful case selection, be it patients or cancer subtypes, for treatments including autophagy inhibition can remedy the undesired effects.

Although the article came on one side of the argument, it presented well balanced review of the issue. In particular, studies on the beneficiary/counterproductive roles of autophagy/autophagy inhibition were presented. Because the article suggests that autophagy inhibition therapies on trials may carry the risk of promoting tumor progression, it would be fitting to present findings, if available, from completed or ongoing trails to support the claim.

Thank you for your comment. Fortunately for cancer patients enrolled in clinical trials in which autophagy is being inhibited for cancer therapy, no counter-productive side effects have been reported. Thus, in order to respond to this comment, we wrote about the current status of clinical trials in the Final remarks section and finished this section with the following paragraph:

Fortunately, no unfavorable outcomes have been reported for the clinical trials using inhibitors of autophagy for the treatment of several types of cancer. With this review, we do not pretend to discourage the use of autophagy inhibitors in cancer clinical trials but rather, discuss evidence in the literature, mostly in vitro but also in vivo, that suggests that autophagy inhibition could be promoting EMT at least in some cases. Thus, it will be central to understand in which cancer types autophagy inhibition could have a counter-productive effect and if blocking autophagy could promote invasion, if this would be particular to a certain cancer type or if this effect is likely to be more evident in early-stage malignancies that retain their epithelial features and have not triggered EMT.

 The risks of autophagy inhibition therapies have been previously suggested and this review is a valuable contribution on the topic especially by attributing them to promoting EMT. These therapies seem to be working in certain contexts. So, the authors may need to comment on as to why this is the case in some but not other cancer types. In addition, alternative explanations rather than the tumor specificity may be discussed.

Thank you for your comment. In the Final remarks section, we discuss more about the cases in which we consider autophagy inhibition could be having counter-productive effects and suggest alternatives for combination therapies according to the revised literature:

So, despite encouraging data from patient clinical trials, the fact that the inhibition of autophagy could be having undesirable effects in some types of cancer and, as we discuss in this manuscript, inducing EMT and promoting invasion and metastasis, at least in some cases, is a worrisome finding. The mechanisms by which autophagy inhibition would induce EMT include EMT-TF degradation by autophagy or the accumulation of p62 due to the inhibition of autophagy and a consequent increase in NF-κB signaling and transcriptional induction of EMT-TF. If these are the main mechanisms by which inhibition of autophagy promotes EMT, it will be necessary to investigate if autophagic degradation of EMT-TF is a general mechanism or if it is only occurring in certain cell types and if accumulation of p62 and its induction of NF-κB signaling occurs in certain cell types but not others when autophagy is inhibited. Another mechanism proposed for the promotion of malignancy mediated by the inhibition of autophagy is the degradation of Notch1 intracellular domain by autophagy [63]. Thus, it would be important to identify those cancers where autophagy is constitutively inhibiting Notch signaling and avoid using autophagy inhibitors or use them in combination with inhibitors of Notch.

An interesting approach for targeting EMT induced by the inhibition of autophagy has been suggested by Wang et al [56], in the best characterized model of autophagy addiction, e.g. RAS-mutated cancer cells. In this study, despite displaying reduced tumor growth, HRasV12 expressing atg5-/- or atg7-/- tumors had increased EMT marker expression, which was reverted using a NF-kB inhibitor. In this regard, many open clinical trials for targeting autophagy-dependent cancers are using RAS/MAPK mutation or activation as biomarkers for autophagy dependency and these findings suggest avoiding CQ or HCQ use as single agents or using them along with NF-kB inhibitor or inhibitors of malignancy-associated signaling pathways that could be activated by the inhibition of autophagy, even in cancers where the best outcomes are expected.

 The authors went into enough details about basic autophagy and EMT concepts. In addition, they used detailed infographics to simplify complex pathways and mechanisms. A brief introduction to the cancer therapies in general or the ones referred and used in combination with autophagy inhibitors would increase the accessibility of the review to the reader.

Thank you for your comment. We included the following sentences in the autophagy in cancer section:

Cancer patients are treated with surgery together with adjuvant or neoadjuvant therapies, which include radiation, cytotoxic chemotherapy, targeted therapies (in case that the oncogenic driver has been identified) or immunogenic therapy. An important role for autophagy has been described for many types of cancer and for the different types of cancer therapies, indicating a promising role for the manipulation of this process in clinical trials. Currently, several clinical trials are trying to inhibit autophagy by using chloroquine (CQ) or hydroxychloroquine (HCQ) alone or in combination with chemotherapy or targeted therapies in several types of cancer [3].

 The authors described autophagy-dependent degradation of some EMT or metastasis-related proteins such as Snail, Twist, NICD1 so on. In addition, they need to discuss in detail these control pathways described in the following references;

- Zada et al. Control of the epithelial-to-mesenchymal transition and cancer metastasis by autophagy-dependent snai1 degradation. Cells. 2019 ;8.129. doi: 10.3390/cells8020129.

- Qiang, L.; He, Y.-Y. Autophagy deficiency stabilizes TWIST1 to promote epithelial- mesenchymal transition. Autophagy 2014, 10, 1864–5, doi:10.4161/auto.32171.

- Gugnoni, M.; Sancisi, V.; Manzotti, G.; Gandolfi, G.; Ciarrocchi, A. Autophagy and epithelial-mesenchymal transition: an intricate interplay in cancer. Cell Death Dis. 2016, 7, e2520, doi:10.1038/cddis.2016.415.

- Qiang, L.; He, Y.-Y. Autophagy deficiency stabilizes TWIST1 to promote epithelial- mesenchymal transition. Autophagy 2014, 10, 1864–5, doi:10.4161/auto.32171.

- Grassi, G.; Di Caprio, G.; Santangelo, L.; Fimia, G. M.; Cozzolino, A. M.; Komatsu, M.; Ippolito, G.; Tripodi, M.; Alonzi, T. Autophagy regulates hepatocyte identity and epithelial-to-mesenchymal and mesenchymal-to-epithelial transitions promoting Snail degradation. Cell Death Dis. 2015, 6, e1880, doi:10.1038/cddis.2015.249

- Gugnoni, M.; Sancisi, V.; Gandolfi, G.; Manzotti, G.; Ragazzi, M.; Giordano, D.; Tamagnini, I. Tigano, M.; Frasoldati, A.; Piana, S.; Ciarrocchi, A. Cadherin-6 promotes EMT and cancer metastasis by restraining autophagy. Oncogene 2017, 36, 667–677. doi:10.1038/onc.2016.237.

- Ahn, J.-S.; Ann, E.-J.; Kim, M.-Y.; Yoon, J.-H.; Lee, H.-J.; Jo, E.-H.; Lee, K.; Lee, J. S.; Park H.-S. Autophagy negatively regulates tumor cell proliferation through phosphorylation. dependent degradation of the Notch1 intracellular domain. Oncotarget 2016, 7, 79047-79063, doi:10.18632/oncotarget.12986.

Thank you for your comment. These studies have been a great addition to the discussion in our review. We included those references in Table 1 and discussed them in the autophagy and its effects on EMT section.

 Minor points

 Each figure legends repeat some sentences described in the main text.

We checked figure legends and eliminated repeated sentences from the main text.

Authors described the effects of cytokines on EMT in many pages, but they did not mention the functional connection of autophagy with those cytokines during cancer progression.

Thank you for your comment. We added the following sentences at the end of Cytokines in the tumor microenvironment and their effects on EMT section (5) and further discussed this point in the autophagy and its effects on EMT section:

Finally, of relevance to the autophagy field, autophagy has been implicated in the secretion of several pro-inflammatory cytokines, particularly those involved in the promotion of EMT. Thus, as will be furthered discussed in the following section, autophagy-mediated secretion and the secretion of cytokines mediated by the inhibition of autophagy are an important link between autophagy and the modulation of EMT.

Please cross check references.

We checked references and corrected the ones that were wrong.

In page 5, the sentence from line 133 to 138 is not clear.

Thank you for you comment. This paragraph has been modified in the revised version:

So, although decreased autophagy in normal cells would induce cellular damage that could lead to malignancy, a functional autophagic pathway is required for oncogenic progression. This has been demonstrated in diverse genetically modified cancer mouse models. So, in a pancreatic ductal adenocarcinoma mouse model with oncogenic Kras, autophagy inhibition increased the frequency of low-grade, pre-malignant pancreatic intraepithelial neoplasia formation but blocked the progression to high grade intraepithelial neoplasia or adenocarcinoma [24]. In a KRas driven model of lung cancer, Atg5 deletion increased hyperplastic tumor foci formation, but decreased progression to adenocarcinomas and signs of malignancy [25]; and in a Kras, p53-/- mouse model of lung cancer, Atg7 deletion altered tumor fate from adenomas to more benign oncocytomas, characterized by the accumulation of defective mitochondria [26].

There are some typos

Thank you for your comment. We double checked the whole manuscript for grammar and typo errors and corrected them in the revised version.

Reviewer 2 Report

This review summarizes some of the recent development in the field of cancer therapy involving the inhibition of autophagy. The literature being covered is broad. Although this is a controversial area, the authors have been the authors’ comments are objective and balanced. This review will be helpful to researchers working in this field.

One minor point: page 15, line444-446, the two sentences should be combined into one.

Author Response

Point-by-point response

This review summarizes some of the recent development in the field of cancer therapy involving the inhibition of autophagy. The literature being covered is broad. Although this is a controversial area, the authors have been the authors’ comments are objective and balanced. This review will be helpful to researchers working in this field.

One minor point: page 15, line444-446, the two sentences should be combined into one.

Thank you for your comment and careful reading of our manuscript. On the revised version of the manuscript, lines 444-446 were replaced by the following paragraph. We hope that it reads better now.

Fortunately, no unfavorable outcomes have been reported for the clinical trials using inhibitors of autophagy for the treatment of several types of cancer. With this review, we do not pretend to discourage the use of autophagy inhibitors in cancer clinical trials but rather, discuss evidence in the literature, mostly in vitro but also in vivo, that suggests that autophagy inhibition could be promoting EMT at least in some cases. Thus, it will be central to understand in which cancer types autophagy inhibition could have a counter-productive effect and if blocking autophagy could promote invasion, if this would be particular to a certain cancer type or if this effect is likely to be more evident in early-stage malignancies that retain their epithelial features and have not triggered EMT.